# Graph Neural Networks for Multivariate Time-Series Forecasting via Learning Hierarchical Spatiotemporal Dependencies

## Abstract

Multivariate time-series forecasting is one of the essential tasks to draw insights from sequential data. Spatiotemporal Graph Neural Networks (STGNNs) have attracted much attention in this field due to their capability to capture the underlying spatiotemporal dependencies. However, current STGNN solutions still fall short of providing trustworthy predictions due to insufficient modeling of the dependencies and dynamics at different levels. In this paper, we propose a graph neural network model for multivariate time-series forecasting via learning hierarchical spatiotemporal dependencies (HSDGNN). Specifically, we organize variables as nodes in a graph while each node serves as a subgraph consisting of the attributes of variables. Then we design two-level convolutions on the hierarchical graph to model the spatial dependencies with different granularities. The changes in graph topologies are also encoded for strengthening dependency modeling across time and spatial dimensions. We test the proposed model on real-world datasets from different domains. The experimental results demonstrate the superiority of HSDGNN over state-of-the-art baselines in terms of prediction accuracy.

## 1 Introduction

With the development of sensor technology, numerous data from real-life activities such as transportation and energy generation have been collected to assist in further analysis and decision-making processes (Li et al., 2023a; Zheng et al., 2022). The numerical data from different sensors are usually recorded chronologically, forming the branch of multivariate time-series data. The traffic flow or the electricity generation within an area is typical data of this kind. To draw insights from such multivariate data, multivariate time-series forecasting has become a vital task that enables planners to act proactively. For instance, grid operators rely on accurate predictions of power generation for power system scheduling and electricity marketing (Zheng et al., 2023).

Various methods have been proposed for multivariate time-series modeling. Classical methods such as Vector Autoregression (VAR) (Zivot & Wang, 2006) and Gaussian process model (GP) (Cunningham et al., 2012) were first proposed for univariate time-series forecasting and then extended to multivariate cases based on the assumption of linear dependencies among variables. However, most multivariate time series possess inherent nonlinear pairwise dependencies, and it becomes a more cumbersome task for these statistical methods considering a large number of variables. With the success of deep learning techniques, deep learning methods have been proven effective in different scientific areas (Alzubaidi et al., 2021), and have dominated the field of time-series forecasting (Masini et al., 2023). Till now, many deep learning methods have been investigated for this problem, including Convolutional Neural Networks (CNN), Recurrent Neural Networks (RNN), and attention-based models (Fan et al., 2023; Wang et al., 2023; Zhang & Yan, 2022).

Compared with the univariate case, the major difficulty of multivariate time-series forecasting lies in capturing the underlying spatiotemporal dependencies in the data. Recently, Graph Neural Networks (GNN) have attracted much attention in the modeling of multivariate data considering their capability in dealing with relational dependencies (Jiang & Luo, 2022). By organizing variables each consisting of multiple attributes as nodes in a graph, GNN allows each node to be aware of others during the information propagation process. With the advance of GNN, Spatio-temporal Graph

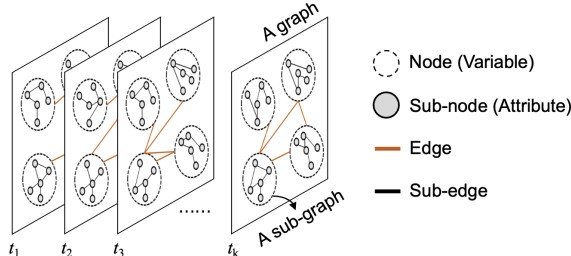

Figure 1: An illustration of the proposed HSDGNN model.

Neural Networks (STGNN) are widely used for multivariate time-series forecasting, in which Graph Convolution Networks (GCN) are applied for spatial modeling, accompanied by sequential models such as RNN (Weng et al., 2023) and Temporal Convolutional Networks (TCN) (Liu et al., 2023) for temporal modeling. Despite the progress that has been made, these models still fall short in multivariate time-series modeling and forecasting due to the following two factors.

First, the dependencies among the attributes of a variable (intra-dependencies for short) are overlooked in current STGNN models. For instance, each traffic signal (e.g., traffic flow, occupancy, and speed) recorded by a detector on the road can be regarded as an attribute of a variable. In typical spatiotemporal forecasting tasks such as traffic flow forecasting, a common strategy to consider the dependency among the traffic flow and other attributes is to project the raw data into a higher dimension via an embedding layer (Liu & Zhang, 2023; Wu et al., 2020). However, following the same transformation operation can not fully exploit the intra-dependencies, which vary with respect to the recording time and the location of detectors. Another popular branch of work only seeks to model the spatial dependencies among the target attributes, i.e. traffic flow of different roads, for a joint prediction. Other related attributes such as traffic speed and road occupancy that are entangled with the prediction target are neglected in the modeling process (Li et al., 2023c; Weng et al., 2023). These methods can provide acceptable results depending on circumstances, but they may also introduce a higher degree of uncertainty, which can impact the reliability of predictions. Second, current STGNN models are ineffective in modeling dynamic spatial dependencies. Most recent works on STGNN replace the traditional pre-defined graph with a series of dynamic graphs derived from data to represent the dynamic relationships among variables (Weng et al., 2023; Li et al., 2023b). Even so, the temporal correlations regarding the changing spatial dependencies are not well-considered, which makes these methods ineffective in learning from dynamic graph topologies.

To bridge these gaps, we propose a graph neural network model for multivariate time-series forecasting via learning hierarchical spatiotemporal dependencies (HSDGNN). As illustrated in Figure 1, a multivariate time series with multiple attributes is first organized as a series of graphs of sub-graphs to explicitly show the hierarchical dependencies inside the data. Herein, variables and attributes are represented by nodes and sub-nodes respectively. The edges connecting different nodes (orange) indicate the spatial dependency among variables while those connecting sub-nodes (black) correspond to the correlations among attributes. In this work, we assume that each variable is comprised of the same set of attributes. Targeting the first issue, we perform an attribute-level sub-graph convolution to capture the time-varying correlations among the attributes of each variable. Then temporal- and spatial-dependency learning processes are leveraged sequentially to facilitate spatiotemporal dependency modeling among variables. Targeting the second issue, we apply an extra temporal learning step to capture the temporal correlations from dynamic graph topologies and generate an enhanced spatiotemporal embedding. After hierarchical dependencies learning, we can generate the predictions by projecting the enhanced spatiotemporal embedding into a desired dimension.

Our contributions can be summarized as follows: **1)** We propose HSDGNN, a hierarchical spatiotemporal dependencies learning based graph neural network model that leverages spatial-, temporal-, and intra-dependency learning in a unified framework. **2)** The temporal correlations among dynamic graph topologies are considered to strengthen dependency modeling across time and spatial dimensions. **3)** We prove the effectiveness of the proposed method on real-world datasets from different domains. The performance improvement when compared to state-of-the-art methods can be up to 15.3% regarding prediction accuracy, without compromising on model size.

## 2 RELATED WORKS

### 2.1 MULTIVARIATE TIME-SERIES FORECASTING

Multivariate time-series forecasting (MTSF) focuses on the modeling and inference of data over time that consists of multiple interdependent attributes. Various methods have been proposed for this problem, from traditional methods to recent deep learning methods. Vector Autoregressive (VAR) (Kilian & Lütkepohl, 2017) and Vector Autoregressive Moving Average (VARMA) (Isufi et al., 2019) are typical extensions of statistical univariate time-series forecasting models for multivariate cases. Gaussian Processes (GP), a non-parametric method for distribution modeling, can also be applied to MSTF tasks (Chen & Sun, 2021). Nevertheless, neither of these traditional methods are competent for the modeling of different forms of nonlinearity in multivariate time series. With the development of deep learning, many deep neural network models have been proposed for MSTF and have proven to outperform traditional methods. LSTNet (Lai et al., 2018) and TPA-LSTM (Shih et al., 2019) are two early deep-learning models that employ CNN and RNN for capturing spatial and temporal dependencies in multivariate time series. Other works replace RNN with convolutional components such as temporal convolution networks (TCN) (Fan et al., 2023) and CNN (Huang et al., 2019) to alleviate the vanishing gradients problem while capturing a sufficient amount of temporal contexts. Empowered by the attention mechanism, many transformer-based models have also been proposed for MSTF (Zhou et al., 2021; 2022; Nie et al., 2023). However, attention-based methods suffer from high computational and memory costs.

### 2.2 GRAPH NEURAL NETWORKS FOR SPATIOTEMPORAL FORECASTING

Multivariate time series can be viewed naturally from the graph perspective considering the complex interconnections among variables. Early works adopt a grid-based graph and assume spatial dependencies can be constructed in the Euclidean space (Pan et al., 2019; Lin et al., 2019). However, the dynamic spatial correlations among variables can not be well approximated in a grid of regular shape. Later works tend to perform message passing in an arbitrary graph to ease the restriction of capturing spatial interactions among different variables. For instance, DCRNN (Li et al., 2018) regards the interactions among variables as a directed diffusion process and utilizes diffusion convolution for message passing. STGCN (Yu et al., 2018) also applies Graph Convolution Networks (GCN) for spatial dependency modeling. However, these works construct the graph based on geographic location information, which is either not provided in many MTSF tasks or insufficient to reflect the genuine dependencies among variables. To reduce the reliance on a user-defined graph structure, many researchers propose to derive an adaptive graph structure from data (Wu et al., 2019; Guo et al., 2019; Bai et al., 2020). Nevertheless, both the user-defined graph and the adaptive graph are static, without considering the potential time-varying spatial topologies. The generation of dynamic graphs has therefore been considered a novel direction for STGNN recently. For example, DGCRN (Li et al., 2023a) integrates the pre-defined graph with dynamic graphs generated with respect to time in the graph convolution module. SDGL (Li et al., 2023c) adopts the static graph as an inductive bias in learning dynamic graphs from node-level inputs. DDGCRN (Weng et al., 2023) combines a dynamic graph embedding with graph signals to derive dynamic graph topologies. Despite the efforts that have been made, none of these methods has considered the intra-dependencies among the attributes of a variable, substantially increasing the risk of giving imprecise predictions. To the best of our knowledge, DMSTGCN (Han et al., 2021) is the only work that explicitly models the effects of additional attributes. However, it compromises model scalability by employing parallel duplicated structures for each additional attribute with a fusion module at the end.

## 3 PROPOSED METHOD

### 3.1 PROBLEM FORMULATION

This paper focuses on the prediction of multivariate time series with multiple attributes. Formally, the observations of $N$ correlated multivariate time series during the past $T$ steps are defined as $\mathbf{X} = \{\boldsymbol{X}_{-T+1}, \boldsymbol{X}_{-T+2}, ..., \boldsymbol{X}_0\}$, where $\boldsymbol{X}_t = \{\boldsymbol{X}_{t,1}, \boldsymbol{X}_{t,2}, ..., \boldsymbol{X}_{t,N}\}$ ($\boldsymbol{X}_{t,j} \in \mathbb{R}^C, t \in [-T+1, 0]$) records the values of $N$ variables with $C$ attributes. We also add the time information for each variable as an extra attribute. Considering the spatiotemporal dependencies among different variables,

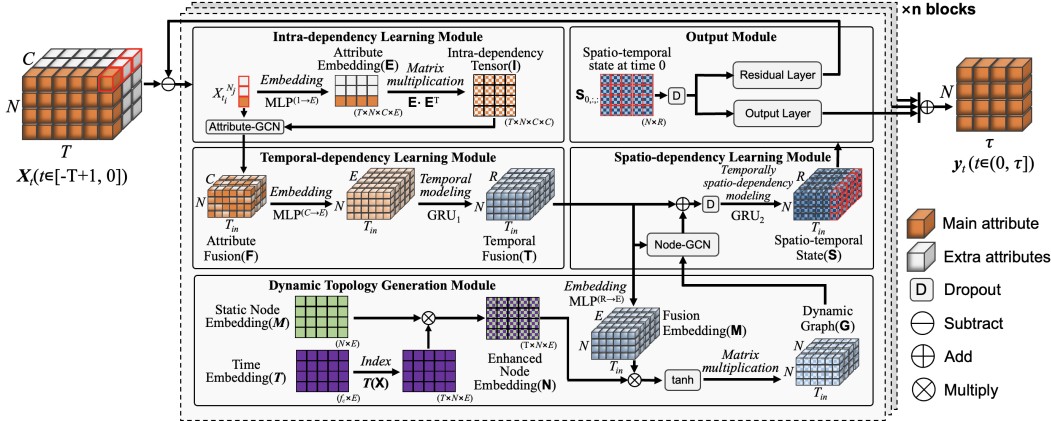

Figure 2: Overview of the proposed HSDGNN model. Herein, the model consists of five modules: Intra-dependency Learning, Temporal-dependency Learning, Dynamic Topology Generation, Spatio-dependency learning, and Output Module.

the variables are organized as a graph $\mathcal{G} = (\mathcal{G}_s, \mathcal{E}, \mathcal{A})$, where $\mathcal{G}_s$ is a set of nodes corresponding to $N$ variables, $\mathcal{E}$ contain the edges, and $\mathcal{A} \in \mathcal{R}^{N \times N}$ is the adjacency matrix indicating the intensity of dependencies among different nodes. Inside each node, multiple attributes of a variable are organized as a sub-graph $\mathcal{G}_s = (\mathcal{V}_f, \mathcal{E}_f, \mathcal{A}_f)$ as they also have spatiotemporal dependencies amongst each other. Herein, $\mathcal{V}_f$, $\mathcal{E}_f$, and $\mathcal{A}_f$ are the sub-nodes corresponding to the $C$ attributes, the sub-edges, and the adjacency matrix among attributes, respectively. Given the observations organized as a hierarchical graph, the target is to find a function $\mathcal{F}$ to jointly predict the values of the main attribute of these variables, which can be formulated as follows:

$$\{\hat{\boldsymbol{y}}_0, \hat{\boldsymbol{y}}_1, ..., \hat{\boldsymbol{y}}_\tau\} = \mathcal{F}(\mathbf{X}|\mathcal{G}, \mathcal{G}_s) \tag{1}$$

where $\hat{\boldsymbol{y}}_i$ ($i \in [0, \tau]$) is the prediction of $N$ variables at timestamp $i$. Taking traffic flow forecasting as an example, we organize all attributes such as traffic flow, speed, and occupancy as a sub-graph and only focus on the prediction of the main attribute (i.e., traffic flow).

## 3.2 MODEL ARCHITECTURE

An overview of HSDGNN is shown in Figure 2, which consists of five components: Intra-dependency Learning, Temporal-dependency Learning, Dynamic topology generation, Spatial-dependency Learning, and Output module. The details of each module are illustrated as follows.

**Intra-dependency Learning.** As shown in previous works, multivariate time-series forecasting considering the auxiliary attributes of a variable can benefit the prediction of the main attribute. A sub-graph convolution is introduced in the Intra-dependency Learning Module to model the time-varying dependencies among multiple variable attributes, which essentially performs an attribute-level graph convolution inside each node. We first embed each attribute of all variables at timestamp $t$ ($t \in [-T + 1, 0]$) using a fully-connected layer:

$$\mathbf{E} = \theta(\boldsymbol{W}_I \boldsymbol{X}_t + \boldsymbol{b}_I) \tag{2}$$

where $\mathbf{E}$ is the attribute embedding, $\theta$, $\boldsymbol{W}$, and $\boldsymbol{b}$ are the activation function, the embedding weight, and the bias, respectively. The subscript $I$ refers to the component belonging to the intra-dependency learning module. We use the same notation hereinafter. Then we can infer the intra-dependencies among multiple attributes by multiplying the attribute embedding $\mathbf{E}$ with $\mathbf{E}^T$:

$$\mathbf{R} = \mathrm{ReLU}(\mathbf{E} \cdot \mathbf{E}^T) \tag{3}$$

where $\mathbf{R}$ is the intra-dependency tensor. GCN is then deployed to capture the intra-dependency inside each variable node. Following (Simonovsky & Komodakis, 2017), we adopt the graph convolution operation in the form of first-order Chebyshev polynomial expansion as follows:

$$\mathbf{F} = (\boldsymbol{I}_f + \boldsymbol{D}^{-\frac{1}{2}} \boldsymbol{A} \boldsymbol{D}^{-\frac{1}{2}}) \mathbf{X} \Theta_I \tag{4}$$

$$= (\boldsymbol{I}_f + \mathbf{R}) \mathbf{X} \Theta_I \tag{5}$$

where $\boldsymbol{I}_f$ is the identity matrix, $\boldsymbol{A}$ and $\boldsymbol{D}$ are the conventional adjacency matrix and the degree matrix, $\mathsf{X}$ and the attribute fusion $\mathsf{F}$ are the input and the output of GCN, $\Theta$ parameterizes the filter. Furthermore, we use the intra-dependency tensor $\mathsf{R}$ as a direct approximation of $\boldsymbol{D}^{-\frac{1}{2}}\boldsymbol{A}\boldsymbol{D}^{-\frac{1}{2}}$ instead of $\boldsymbol{A}$, thus avoiding the computational cost when calculating the Laplacian matrix $\boldsymbol{D}$.

**Temporal-dependency Learning.** Given the attribute fusion that encodes rich information from auxiliary attributes, a temporal-dependency learning module is applied to capture the temporal patterns of each variable. Likewise, we use a fully connected layer for embedding the attribute fusion and a GRU for temporal modeling, formulated as follows:

$$\boldsymbol{h}_{G_1}^t = \mathrm{GRU}_1(\theta(\boldsymbol{W}_T \cdot \mathsf{F} + \boldsymbol{b}_T), \boldsymbol{h}_{G_1}^{t-1} \mid \boldsymbol{W}_{G_1}), \tag{6}$$

where $\boldsymbol{h}_{G_1}^{t-1}$ is initialized as zero. The temporal fusion $\mathsf{T}$ is then comprised of the hidden states $\boldsymbol{h}_{G_1}^t$ ($t \in [-T+1, 0]$), which capture the sequential patterns of different attributes. By incorporating relevant information along the temporal dimension, this process also attends to the temporal impact of the auxiliary attributes on the main attribute.

**Dynamic topology generation.** Defining the graph topology is a prerequisite for deploying GCN to capture the spatial dependencies among variables. However, a pre-defined or static topology is ill-suited for defining the varying spatial relationships. Following recent works, we propose to derive dynamic spatial topology along with the model training process in an end-to-end manner. The fundamental difference is that we also highlight the importance of the intra-dependencies among attributes by generating hierarchical dynamic graph topologies.

Specifically, we first initialize a node-embedding matrix ($\boldsymbol{M}_e \in \mathbb{R}^{N \times E}$), where each row represents the unique embedding of node $j$ ($j \in [1, N]$) and $E$ is the embedding dimension. Inspired by (Han et al., 2021; Weng et al., 2023), we initialize another time-embedding matrix $\boldsymbol{T}_e \in \mathbb{R}^{f_c \times E}$ to incorporate the time information, where the number of rows $f_c$ depends on the sampling rate of variables. We then use $\mathsf{X}$ as the index to retrieve time information for each timestamp from $\boldsymbol{T}_e$. In what follows, an element-wise multiplication is performed between the retrieved information and the node-embedding to derive the enhanced node-embedding matrix $\mathsf{N}_e = \boldsymbol{M}_e \odot \boldsymbol{T}_e(\mathsf{X})$. We then perform another element-wise production to update $\boldsymbol{N}_e$ with the fusion embedding $\mathsf{M}$ embedded from the dynamic graph signals. After that, we generate the dynamic spatial topology by calculating the similarity among the updated node embeddings. These steps can be formulated as follows:

$$\mathsf{M} = \theta(\boldsymbol{W}_D \cdot \mathsf{T} + \boldsymbol{b}_D) \tag{7}$$

$$\mathsf{G} = \mathrm{ReLU}(\tanh(\mathsf{M} \odot \mathsf{N}_e) \cdot \tanh(\mathsf{M} \odot \mathsf{N}_e)^{\mathrm{T}}) \tag{8}$$

Herein, the state transition tensor $\mathsf{G}$ is adopted to directly approximate the state transition process (Li et al., 2018) on the graph. The derived state transition tensor $\mathsf{G}$ accompanied by the intra-dependency tensor $\mathsf{R}$ enables convolutions on a hierarchical graph without relying on a fixed graph topology.

**Spatial-dependency Learning.** Given the dynamic graph topology $\mathsf{G}$, the graph signal $\mathsf{T}$ can be organized and processed according to their dynamic spatial relationships. We adopt the diffusion convolution (Li et al., 2018) for information aggregation on the graph considering computational efficiency. This process can be mathematically realized by multiplying the transition matrices with the graph signals and learnable parameters as follows:

$$\mathsf{Z} = \sum_{k=0}^{K} \mathsf{G}^k \cdot \mathsf{T} \cdot \Theta_S^k \tag{9}$$

where $\mathsf{Z}$ denotes the output, $K$ is the maximum diffusion steps, $\Theta_S \in \mathbb{R}^{N \times R \times D_z}$ parameterizes the filter, $R$ is the hidden state dimension of GRU1, $D_z$ is the output dimension. We apply the dynamic graph topology $\mathsf{G}$ as the approximation of the original transition matrix.

Furthermore, we adopt node adaptive parameter learning (Bai et al., 2020) to learn node-specific patterns, where the shared parameter space $\Theta_S$ is factorized into the node embedding $M_e$ and learnable matrices. The final spatial-dependency learning process can be formulated as follows:

$$\mathsf{Z} = \sum_{k=0}^{K} \mathsf{G}^k \cdot \mathsf{T} \cdot M_e \cdot \boldsymbol{W}_S^k + M_e \cdot \boldsymbol{b}_S^k \tag{10}$$

Different from current STGNN approaches which only model spatial and temporal dependency in separate modules, we apply an extra temporal learning component in our spatial-dependency learning module to consider the change of graph topology. This is essential for strengthening dependency modeling across time and spatial dimensions. Specifically, the graph signal after the diffusion process will serve as the input of a second GRU component for temporally spatial-dependency modeling, which can be formulated as follows

$$\boldsymbol{h}_{G_2}^t = \mathrm{GRU}_2(\mathrm{D}(\mathbf{Z})), \boldsymbol{h}_{G_2}^{t-1} \mid \boldsymbol{W}_{G_2}), \tag{11}$$

where D is the dropout operation, $\boldsymbol{h}_{G_2}^t$ is the enhanced spatiotemporal embedding.

**Output Module.** The output module consists of two linear layers. The $l$-th output layer generates the predictions $\hat{\boldsymbol{Y}}_t^l$ given the enhanced spatiotemporal embedding $\boldsymbol{h}_{G_2}^0$ at $t = 0$. The residual layer resembles the structure of the output layer but is applied to reconstruct the variable signal $\boldsymbol{X}$. The residual part which is a portion of the variable signals that cannot be approximated well by the current block will be modeled by the next block. Inspired by residual learning (He et al., 2016), the residual between the original signals and the predictions is minimized by this backward branch, which eases the training of deeper networks as we stack several blocks to enhance the model's learning capability. The outputs of all blocks are summed up to generate the final predictions.

Different from recent STGNN models (Li et al., 2023a; Weng et al., 2023), we decouple the generation and updating of dynamic graph topologies from the temporal-dependency learning process. Although modeling dependencies at different levels may incur high computational demand, we will prove in the experiment that HSDGNN can benefit from the decoupled scheme and perform efficiently compared to state-of-the-art GNN methods.

## 4 EXPERIMENTS

In this section, we present a set of experiments to evaluate the performance of the proposed model from both effectiveness and efficiency perspectives. The code to reproduce the experiments is available in a repository [1].

### 4.1 EXPERIMENT SETUP

**Benchmark Datasets.** For our evaluations, we require multivariate time series with multiple attributes involving different spatiotemporal dependencies. Thus we have chosen two widely used real-world traffic datasets from the literature (Guo et al., 2019), namely PEMSD4 and PEMSD8, which consist of three kinds of traffic measurements. The raw data are collected by the Caltrans Performance Measurement System (PeMS) [2] every 30 seconds and aggregated within consecutive 5-minute intervals. The authors (Guo et al., 2019) adopt several strategies for data preprocessing, including removing redundant detectors, filling the missing values, and normalizing the data. Following the same steps, we generate two datasets from the PeMS to complement our experiment, denoted as PEMSD5 and PEMSD11 corresponding to the data records from districts 5 and 11, respectively. To encourage a comprehensive evaluation of our model on different applications, we also conduct experiments on an open-access electricity time-series dataset (Zheng et al., 2022), which contains minute-level load and renewable energy over 3 years across the US. The details of these datasets can be found in Appendix A.1. All benchmark datasets are also released in the repository.

**Baseline Methods.** We compare the proposed model with both traditional methods and state-of-the-art deep learning (DL) methods, including History Average Model (HA) (Hamilton, 2020), Autoregression (AR) (Box et al., 2015), LSTNet (Lai et al., 2018), AGCRN (Bai et al., 2020), ST-AE (Liu et al., 2023), SDGL (Li et al., 2023c), and DDGCRN (Weng et al., 2023). Herein, HA and AR are traditional methods that assume pre-defined structural patterns in data. LSTNet is an early deep-learning method relying on CNN for spatial dependency learning. Among the other STGNN-based methods, AGCRN and ST-AE both derive an adaptive static graph to reduce the reliance on a pre-defined graph structure. SDGL and DDGCRN are more recent works that generate dynamic graph topologies for spatial dependency modeling while DDGCRN is considered state-of-the-art.

---

[1] https://anonymous.4open.science/r/HSDGNN-A

[2] https://pems.dot.ca.gov

**Performance Metrics.** The proposed HSDGNN method is evaluated from three perspectives. First, we adopt commonly used metrics to quantify the prediction accuracy, including Mean Absolute Error (MAE), Root Mean Square Error (RMSE), and Mean Absolute Percentage Error (MAPE). Second, we compare the number of model parameters of different methods to prove the advantage of HSDGNN regarding model complexity. To quantitively show the efficiency of different methods, we include the actual model running time during the training and inference stages as the third metric.

## 4.2 MAIN RESULTS

Table 1 summarizes the prediction results of all baselines on the 5 benchmark datasets. For each dataset, we train the baselines 10 times and report the average values of the three performance metrics with standard deviations. The implementation details can be found in Appendix A.1. As can be seen, the proposed HSDGNN method shows state-of-the-art performance on all datasets. Compared to DDGCRN, HSDGNN can achieve a performance improvement of up to 11.8%, 15.3%, and 9.8% in terms of the three performance metrics. We can draw the main conclusions from Table 1 as follows: 1) DL methods outperform traditional methods in general since the nonlinearity in multiple variables is not well-modeled by statistical approaches (others *versus* HA and AR); 2) GNN is more effective than CNN in capturing spatial dependencies among variables, given higher prediction accuracy (GNN-based methods *versus* LSTNet); 3) Adopting dynamic graph topologies other than a fixed adjacency matrix learned from data improves the performance of GNN-based models (HSDGNN, DDGCRN, and SDGL *versus* AGCRN and ST-AE); 4) Incorporating time information contributes to the identification of time-related patterns and thus eases the training of models (HSDGNN and DDGCRN *versus* others); 5) HSDGNN provides more promising prediction results by explicitly considering the intra-dependencies among the attributes of each variable (HSDGNN *versus* others).

Table 1: Comparions of the performance of all baselines on the 5 benchmark datasets. We report the average performance of 10 runs with standard deviations in parentheses. The best and the second-best results in each case are marked in bold and underlined, respectively. Note that the results on the PSML datasets are $10^3$ times larger than the actual values for better readability.

| METHODS | PEMSD4 | | | PEMSD5 | | | PEMSD8 | | | PEMSD11 | | | PSML | | |
|---|---|---|---|---|---|---|---|---|---|---|---|---|---|---|---|
| | MAE | RMSE | MAPE (%) | MAE | RMSE | MAPE (%) | MAE | RMSE | MAPE (%) | MAE | RMSE | MAPE (%) | MAE | RMSE | MAPE (%) |
| HA | 35.173 - | 52.346 - | 25.453 - | 17.818 - | 27.446 - | 26.541 - | 29.189 - | 43.540 - | 18.448 - | 29.861 - | 45.819 - | 27.769 - | 5.104 - | 7.004 - | 513.017 - |
| AR | 27.827 - | 43.930 - | 18.973 - | 14.394 - | 22.738 - | 21.481 - | 22.385 - | 35.456 - | 14.622 - | 21.879 - | 35.477 - | 21.259 - | 0.597 - | 1.380 - | 61.250 - |
| LSTNet | 19.932 (±0.123) | 31.564 (±0.132) | 14.016 (±0.143) | 12.004 (±0.067) | 17.810 (±0.109) | 22.805 (±0.854) | 16.601 (±0.238) | 25.817 (±0.289) | 10.729 (±0.285) | 14.657 (±0.046) | 23.743 (±0.058) | 16.180 (±0.174) | 0.477 (±0.066) | 1.325 (±0.073) | 48.156 (±6.426) |
| AGCRN | 19.316 (±0.054) | 31.561 (±0.170) | 12.871 (±0.070) | 11.413 (±0.153) | 17.598 (±0.297) | 20.309 (±0.491) | 15.837 (±0.165) | 25.219 (±0.228) | 10.316 (±0.135) | 14.217 (±0.040) | 23.526 (±0.137) | 14.676 (±0.115) | 0.396 (±0.025) | 1.024 (±0.029) | 41.239 (±2.564) |
| ST-AE | 19.908 (±0.083) | 31.257 (±0.093) | 13.896 (±0.327) | 11.186 (±0.151) | 17.127 (±0.288) | 16.317 (±0.391) | 15.960 (±0.165) | 24.877 (±0.214) | 10.220 (±0.175) | 15.108 (±0.149) | 24.219 (±0.206) | 15.002 (±0.292) | 0.435 (±0.042) | 1.046 (±0.040) | 45.025 (±4.481) |
| SDGL | 18.625 (±0.063) | 31.069 (±0.236) | 12.400 (±0.088) | 10.786 (±0.112) | 16.810 (±0.216) | 17.150 (±0.577) | 14.944 (±0.059) | 24.166 (±0.110) | 9.597 (±0.085) | 13.705 (±0.017) | 22.656 (±0.066) | 13.796 (±0.061) | 0.330 (±0.037) | 0.943 (±0.018) | 33.22 (±3.207) |
| DDGCRN | 18.460 (±0.093) | 30.864 (±0.324) | 12.290 (±0.134) | 10.802 (±0.085) | 16.898 (±0.218) | 16.979 (±0.433) | 14.382 (±0.064) | 23.793 (±0.166) | 9.446 (±0.088) | 13.754 (±0.035) | 22.849 (±0.105) | 13.985 (±0.159) | 0.278 (±0.027) | 0.766 (±0.033) | 28.431 (±2.829) |
| HSDGNN | **18.254** (±0.025) | **30.443** (±0.132) | **12.065** (±0.177) | **10.414** (±0.115) | **16.526** (±0.313) | **15.387** (±0.286) | **13.718** (±0.154) | **23.405** (±0.137) | **9.077** (±0.136) | **13.586** (±0.025) | **22.627** (±0.058) | **13.706** (±0.086) | **0.245** (±0.023) | **0.649** (±0.067) | **25.632** (±3.740) |

To draw insights from these results, we visualize the stepwise performance of different methods in Figure 3, where HA and AR are not considered as they generate deterministic results. Due to the page length limit, we only show the comparison on the PEMSD4 dataset (See Appendix A.2 for the comparisons on other datasets). We can see that HSDGNN achieves consistent improvements over other baselines regarding different prediction horizons. The relatively small deviations in the results given by HSDGNN also prove the stability of our model under different initialization conditions.

In Table 2, we compare the complexity and efficiency of HSDGNN against other DL baselines in terms of the number of model parameters and actual running time during training and testing stages. The comparisons are based on the default settings of all methods with the same batch size of 64. Regarding model complexity, all baselines have comparable model sizes on different datasets despite the ST-AE on PEMSD4, which contains more variables than other datasets. Furthermore, ST-AE and SDGL may encounter scalability issues since these models' parameters grow with a power larger than 1 regarding the number of variables. A quantitative analysis of such issues can be found in Appendix A.2. Given large model sizes, these methods are ill-suited for a resource-constrained environment. On the contrary, the model size of HSDGNN stays nearly constant in different cases,

demonstrating better model scalability. Regarding the actual running time, LSTNet can be executed fastest with a pure CNN structure but at the expense of prediction accuracy. HSDGNN ranks third among the other GNN-based methods, being less efficient than SDGL and AGCRN. However, AGCRN achieves inferior performance without considering the time-varying spatial dependencies among variables while SDGL suffers from the scalability issue. As a result, we can conclude that HSDGNN achieves a better trade-off between model performance and computing resources.

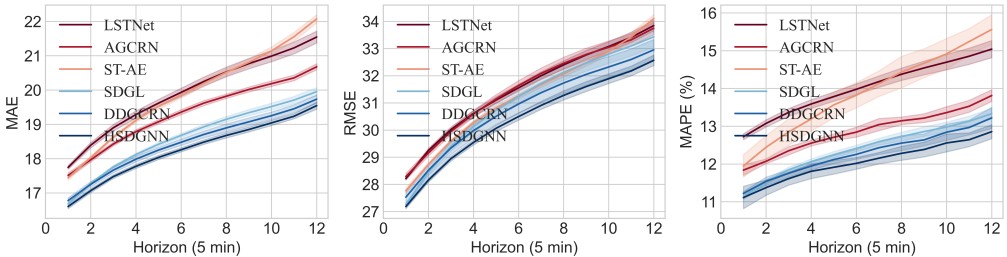

Figure 3: The stepwise performance of different models on the PEMSD4 dataset. Here we show the average performance of 10 runs with the standard deviation for each model.

Table 2: Comparisons of all DL baselines regarding efficiency and model size.

| METHODS | DATASETS | | | | | | | | | | | | | | |
| | PEMSD4 | | | PEMSD5 | | | PEMSD8 | | | PEMSD11 | | | PSML | | |
| | Time (s/epoch) | | Params | Time (s/epoch) | | Params | Time (s/epoch) | | Params | Time (s/epoch) | | Params | Time (s/epoch) | | Params |
| | Train | Test | (M) | Train | Test | (M) | Train | Test | (M) | Train | Test | (M) | Train | Test | (M) |
| LSTNet | 0.891 | 0.174 | 0.617 | 0.449 | 0.069 | 0.190 | 0.561 | 0.091 | 0.369 | 0.727 | 0.125 | 0.394 | 4.428 | 1.419 | 0.181 |
| AGCRN | 13.907 | 1.732 | 0.749 | 9.390 | 1.156 | 0.746 | 9.940 | 1.331 | 0.747 | 10.348 | 1.425 | 0.748 | 22.455 | 2.878 | 0.746 |
| ST-AE | 57.859 | 14.297 | 3.190 | 49.629 | 13.588 | 0.320 | 54.387 | 14.639 | 1.090 | 60.062 | 16.436 | 1.249 | 141.983 | 36.664 | 0.298 |
| SDGL | 8.788 | 0.895 | 0.998 | 3.516 | 0.371 | 0.492 | 5.229 | 0.514 | 0.677 | 5.175 | 0.484 | 0.706 | 9.722 | 0.886 | 0.483 |
| DDGCRN | 43.533 | 4.062 | 0.569 | 20.253 | 1.969 | 0.565 | 24.626 | 2.286 | 0.567 | 24.987 | 2.436 | 0.567 | 48.614 | 4.840 | 0.576 |
| HSDGNN | 33.526 | 2.428 | 0.583 | 7.974 | 0.662 | 0.576 | 18.864 | 1.509 | 0.579 | 20.056 | 1.481 | 0.579 | 19.511 | 1.793 | 0.612 |

## 4.3 ABLATION STUDIES

In this section, we conduct ablation studies to validate the effectiveness of different components in HSDGNN. We have considered HSDGNN without the following components: 1) **HSDGNN_w/o_IDLM**: HSDGNN without the intra-dependency learning module. We omit this module to demonstrate the necessity of intra-dependency modeling for MTSF. 2) **HSDGNN_w/o_MF**: HSDGNN with the input containing only the main attribute. Through this setting, we can directly prove the superiority of the proposed model architecture against other baselines given the same input. 3) **HSDGNN_w/o_GRU$_1$**: HSDGNN without the first GRU component in the temporal-dependency learning module. 4) **HSDGNN_w/o_GRU$_2$**: HSDGNN without the second GRU component in the spatial-dependency learning module. Herein, we verify the indispensability of modeling the change of graph topologies. 5) **HSDGNN_w/o_DG**: HSDGNN without dynamic graph.

The performances of different variants are summarized in Table 3, which we can conclude as follows: 1) The completed model outperforms all variants on each dataset, verifying the necessity of different components. 2) HSDGNN_w/o_IDLM performs slightly better than HSDGNN_w/o_MF but obviously worse than the completed model. It proves that considering multiple inter-dependent attributes can only provide a limited improvement regarding the prediction accuracy without effectively modeling the intra-dependencies among the attributes. Besides, HSDGNN_w/o_MF still outperforms DDGCRN, which shows the superiority of the proposed model architecture. 3) The exclusion of GRU$_2$ can negatively affect our model's ability more than that of GRU$_1$. This demonstrates the different focuses of the two GRU components and the essentiality of temporally spatial-dependency modeling. 4) HSDGNN_w/o_DG outperforms the GNN-based baselines relying on a static graph, which indicates that learning hierarchical spatiotemporal dependencies from data contributes to the performance of MSTF models.

Table 3: The results of ablation studies on HSDGNN.

| METHODS | DATASETS | | | | | | | | | | | | | | |
| | PEMSD4 | | | PEMSD5 | | | PEMSD8 | | | PEMSD11 | | | PSML | | |
| | MAE | RMSE | MAPE (%) | MAE | RMSE | MAPE (%) | MAE | RMSE | MAPE (%) | MAE | RMSE | MAPE (%) | MAE | RMSE | MAPE (%) |
|---|---|---|---|---|---|---|---|---|---|---|---|---|---|---|---|
| HSDGNN | **18.254** | **30.443** | **12.065** | **10.414** | **16.526** | **15.387** | **13.718** | **23.405** | **9.077** | **13.586** | **22.627** | **13.706** | **0.245** | **0.649** | **25.632** |
| HSDGNN_w/o_IDLM | 18.276 | 30.672 | 12.088 | 10.456 | 16.586 | 15.829 | 13.756 | 23.490 | 9.128 | 13.599 | 22.677 | 13.799 | 0.252 | 0.657 | 26.325 |
| HSDGNN_w/o_MF | 18.277 | 30.605 | 12.098 | 10.547 | 16.942 | 16.293 | 14.060 | 23.611 | 9.158 | 13.631 | 22.701 | 13.713 | 0.256 | 0.665 | 26.681 |
| HSDGNN_w/o_GRU1 | 18.304 | 30.493 | 12.258 | 10.477 | 16.823 | 15.427 | 13.737 | 23.469 | 9.110 | 13.627 | 22.656 | 13.843 | 0.254 | 0.655 | 26.251 |
| HSDGNN_w/o_GRU2 | 18.558 | 30.636 | 12.216 | 10.509 | 16.506 | 15.786 | 14.366 | 23.564 | 9.366 | 13.711 | 22.758 | 13.858 | 0.323 | 0.767 | 33.306 |
| HSDGNN_w/o_DG | 18.799 | 31.360 | 12.496 | 10.791 | 16.882 | 17.109 | 14.740 | 24.094 | 9.660 | 13.897 | 22.934 | 14.194 | 0.335 | 1.012 | 34.589 |

## 4.4 EFFECT OF HYPERPARAMETERS

In this section, we prove the robustness of HSDGNN against the selection of hyperparameters. We have considered four main hyperparameters of the model: 1) The embedding dimension $E$ used throughout different modules; 2) The hidden state dimension $R$ of both GRU components; 3) The number of blocks $n$; 4) The number of the maximum diffusion step $K$. We illustrate the performance of HSDGNN on the PEMSD8 dataset regarding different hyperparameter settings in Figure 4, where we also include the results given by DDGCRN for comparisons. Given a generally stable performance, we can conclude from the results that HSDGNN is not sensitive to the choice of hyperparameters. Specifically, the prediction accuracy of HSDGNN can be slightly improved with the increase of $E$, $R$, or $n$ before reaching its best performance. On the other hand, increasing the number of diffusion times cannot contribute to a better performance. We can also see that HSDGNN with diverse hyperparameter settings can outperform DDGCRN, despite a few extreme cases.

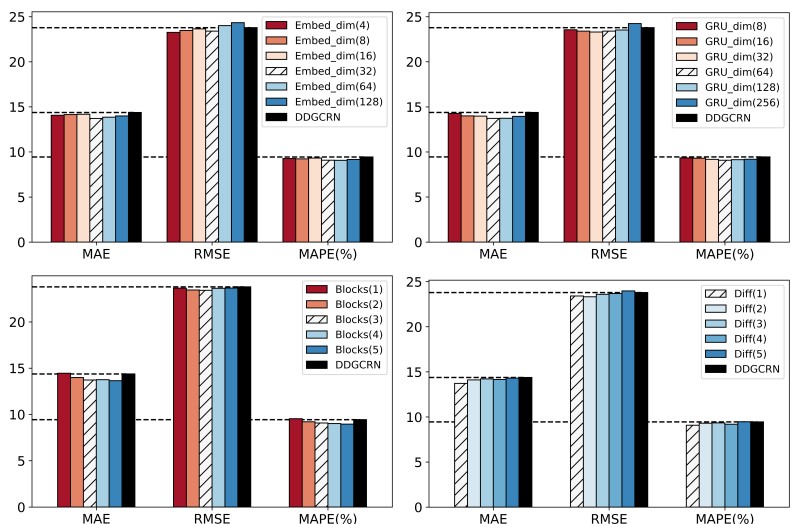

Figure 4: The performance of HSDGNN regarding different hyperparameter settings. The parameter configurations applied in the main experiment are represented by white boxes with slash.

## 5 CONCLUSION AND FUTURE WORK

In this paper, we propose a graph neural network model for multivariate time-series forecasting via learning hierarchical spatiotemporal dependencies (HSDGNN). It organizes multivariate time series with multiple attributes as a hierarchical graph representation and performs two levels of graph convolutions to model the dependencies among variables and attributes. The changes in graph topologies are encoded for strengthening dependency modeling across time and spatial dimensions. The experimental results demonstrate the superiority of HSDGNN over state-of-the-art baselines in terms of prediction accuracy. It has also been proven that HSDGNN is robust towards the choice of hyperparameters. However, we note that modeling the dependencies at different levels can incur relatively high computational demand. In future work, we plan to derive a lightweight network architecture for hierarchical dependency learning, without compromising on model performance.

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

## A APPENDIX

### A.1 SETUP DETAILS

**Benchmark datasets.** We use five datasets from different domains in our experiment. The details of these benchmark datasets are given in Table 4. We also visualize two datasets in Figure 5, from which we can observe obvious correlations among different attributes.

Table 4: The details of the benchmark datasets used in the experiment.

| DATASETS | | SAMPLES | NODES | SAMPLE RATE | ATTRIBUTES |
|---|---|---|---|---|---|
| Traffic | PEMSD4 | 16992 | 307 | 5 min | Flow, Occupancy, Speed |
| | PEMSD5 | 16992 | 71 | 5 min | Flow, Occupancy, Speed |
| | PEMSD8 | 17856 | 170 | 5 min | Flow, Occupancy, Speed |
| | PEMSD11 | 17568 | 184 | 5 min | Flow, Occupancy, Speed |
| Electricity | PSML | 44640 | 66 | 1 min | Load, Wind power, Solar power |

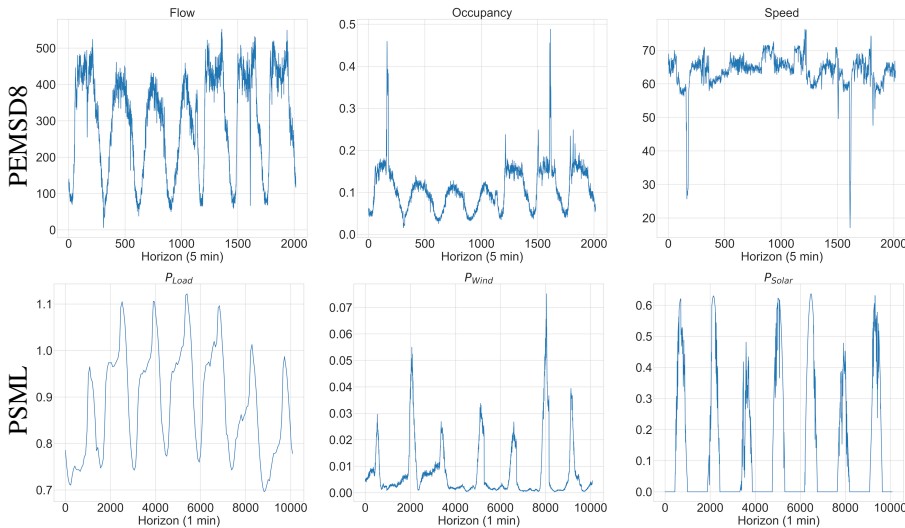

Figure 5: Visualization of the PEMSD8 and PSML datasets.

**Implementation Details.** In this work, we aim to achieve accurate short-term multivariate time-series forecasting. Thus the horizons of observations and predictions are both set to 12, which is in line with previous works. Each dataset has been divided into the training part, the validation part, and the testing part, following the ratio of 60%, 20%, and 20%. Regarding the hyperparameters of HSDGNN, the embedding dim $E$ and the hidden state dimension of both GRU components are set

to 32 and 64, respectively. The maximum diffusion step $K$ is set to 1 for the diffusion convolution. Finally, we stack 3 blocks together to further improve model performance. We train the model with a batch size of 64 for 300 epochs with an early stop patience of 15. The learning rate for the Adam optimizer is initialized to $10^{-3}$ with a decay rate of 0.99 per epoch. For all the baselines, we adopt the default hyperparameter settings while carefully tuning the learning rate on each dataset if required. All experiments are performed on NVIDIA A800 Tensor Core GPUs.

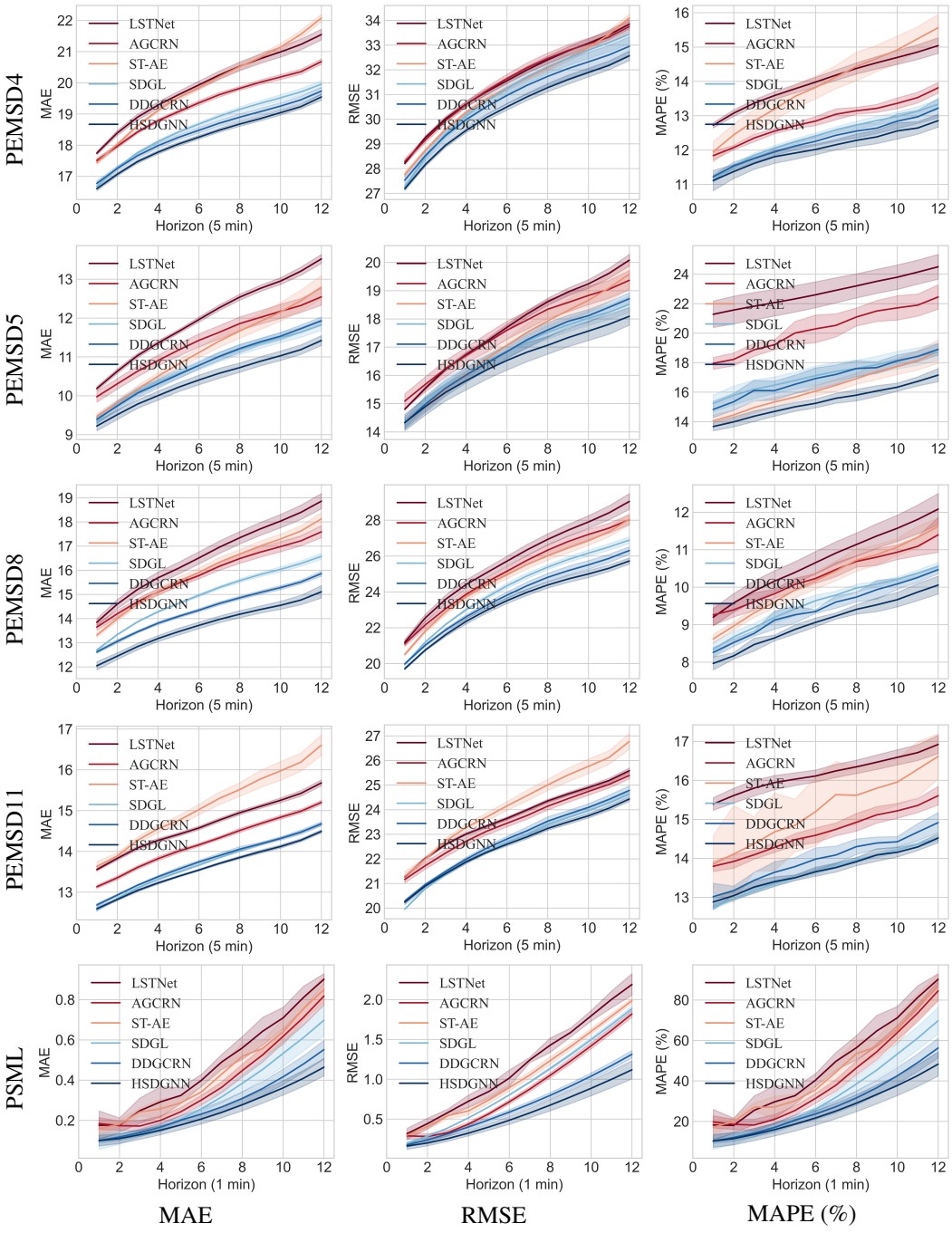

Figure 6: The stepwise performance of different models on all datasets. Here we show the average performance of 10 runs with the standard deviation for each model. We can see that HSDGNN outperforms other baselines in all cases. The superiority is also consistent regarding different prediction horizons with small variations.

## A.2 EXPERIMENT DETAILS

**Stepwise performance.** We only show in Figure 3 the stepwise performance of different methods on the PEMSD4 datasets due to the page length limit. The results on other datasets are shown in Figure 6. We can see that MSDGNN outperforms other baselines regarding different prediction horizons in all five cases. The superiority is also consistent given small variations.

**Long-term prediction performance.** So far we have proved the effectiveness of MSDGNN on short-range forecasting tasks, considering a forecasting horizon of fewer than 60 minutes (12 steps). In this section, we explore the potential of MSDGNN to generate relatively long-term predictions of up to 360 minutes (72 steps). For example, we compare the long-range stepwise prediction performances of MSDGNN and DDGCRN on the PEMSD8 dataset in Figure 7. Owing to the modeling of dependencies and dynamics at different levels, HSDGNN performs consistently better than DDGCRN in providing both short-term and long-term predictions. Besides, HSDGNN is also more stable given a smaller performance variation, especially in terms of MAPE.

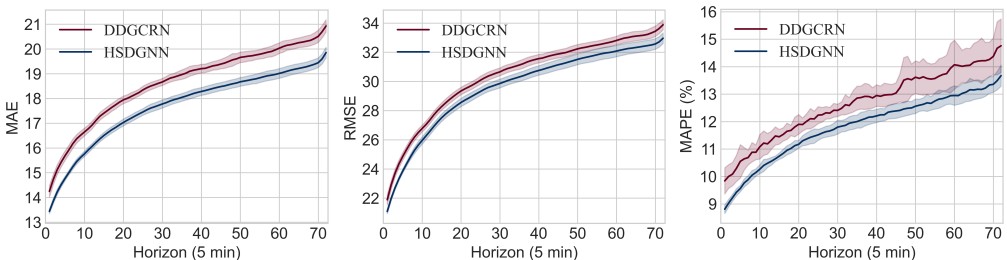

Figure 7: The relatively long-range stepwise performances of HSDGNN and DDGCRN on the PEMSD8 dataset. Here we show the average performance of 30 runs with the standard deviation for each model. We can observe the consistent superiority of MSDGNN over DDGCRN in providing both short-range and long-range predictions.

**Model size regarding the number of nodes.** In previous experiments, we have demonstrated better model scalability of HSDGNN given a consistent model size regarding the number of nodes (variables). In this section, we further conduct a quantitative analysis detailing the advantages of HS-DGNN from this perspective. In Figure 8, we compare the model sizes of different baselines regarding the input data size. Specifically, we choose the PEMSD4 dataset and manually increase the number of nodes by duplicating the original ones several times (from 2 to 32). The model size of each baseline is normalized by dividing the results on original PEMSD4 datasets. Due to the broadly varied range of values, we report the sizes of each model with respect to different numbers of nodes on a logarithmic scale.

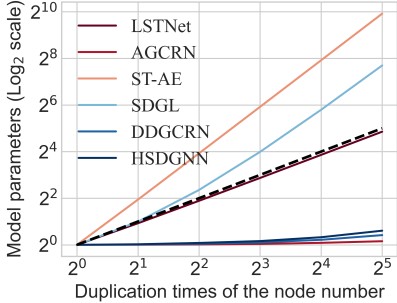

Figure 8: The model size of all DL baselines regarding different numbers of nodes (variables). The dotted line indicates a linear expansion of the model size. We can see that the model size of HSDGNN stays nearly constant when increasing the size of the input data.

As illustrated in Figure 8, ST-AE and SDGL encounter scalability issues since the model size grows with a power larger than 1 regarding the number of nodes. Therefore, they are not suitable for a

resource-constrained environment. The model size of LSTNet grows linearly with the input size, however, the pure CNN architecture is insufficient in modeling spatial dependencies at different levels. On the other hand, AGCRN, DDGCRN, and HSDGNN are insensitive to the size of input data with a nearly constant model size. Nevertheless, AGCRN relies on a fixed graph which can not reflect the dynamic dependencies among different variables. As it has been previously shown in Table 2, HSDGNN is twice more efficient than DDGCRN in terms of training speed. Thus we can demonstrate that HSDGNN outperforms other baselines by trading off prediction performance and computing resources.

**Visualization of hierarchical spatial dependencies.** Compared to a univariate time series, the complexity of a multivariate time series mainly consists in the spatial dependencies that existed at different levels, i.e., among both variables and attributes. In HSDGNN, we design a hierarchical spatiotemporal learning framework to model such spatial dependencies simultaneously. We illustrate the existence of hierarchical spatiotemporal dependencies by visualizing the dynamic graph topologies derived by HSDGNN in Figure 9, where we also include a static graph topology learned by AGCRN for comparison. We can see that the dependencies among both variables and attributes vary with different timestamps. Either overlooking or neglecting these hierarchical spatiotemporal dependencies can damage model performance by giving imprecise predictions.

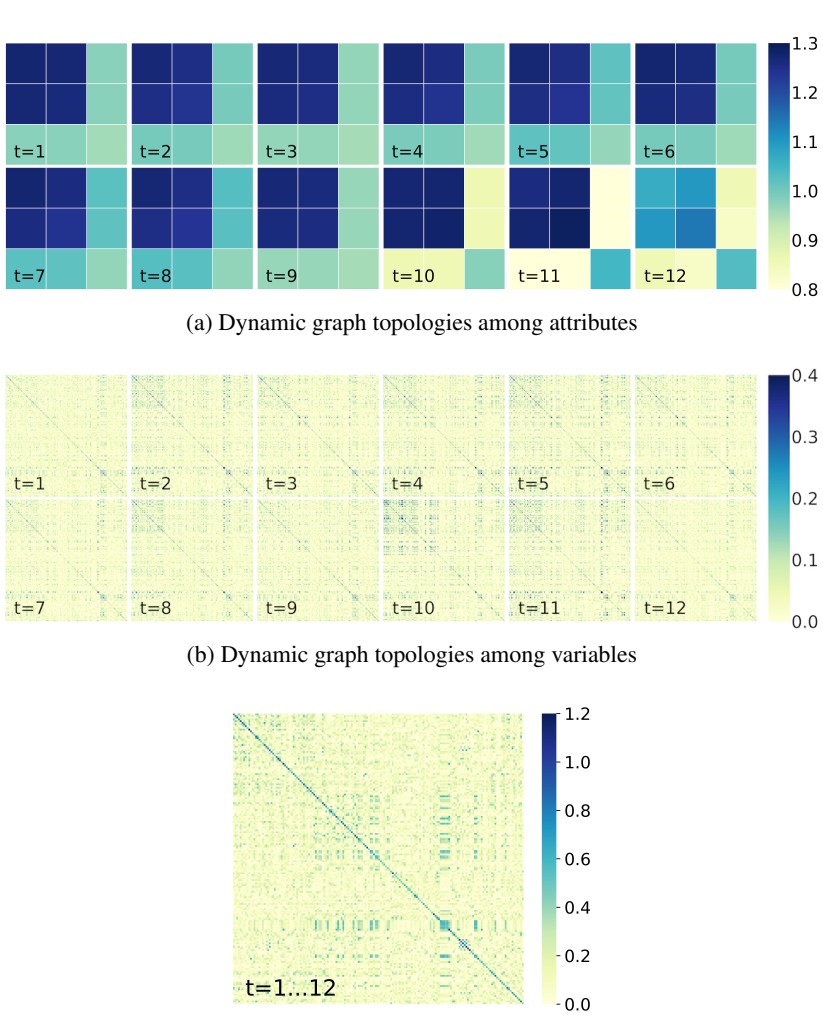

(a) Dynamic graph topologies among attributes

(b) Dynamic graph topologies among variables

(c) A static graph topology

Figure 9: Illustration of hierarchical spatiotemporal dependencies in multivariate time series: (a) Dynamic graph topologies among attributes and (b) Dynamic graph topologies among variables. A static graph structure is visualized in (c).

**Visualization of prediction results.** In this section, we further visualize the prediction results of HSDGNN to provide an intuitive explanation of the superiority of HSDGNN over the state-of-the-art baseline. In Figure 10, we compare the predictions given by HSDGNN and DDGCRN on the PSML dataset as an illustration. We show the results from both the global (left) and local (right) perspectives. We can see that both methods can model the general temporal patterns of the original time series. However, DDGCRN fails to capture the trend in the time series when the underlying behavior of the corresponding signal suddenly changes. HSDGNN achieves better results owing to the sufficient modeling of intra-dependencies among the attributes of a variable, since we can infer the tendency of the main signal from the related signals. Besides, we also consider the temporal correlations among dynamic graph topologies to strengthen the dependency modeling across time and spatial dimensions, and hence a more precise prediction.

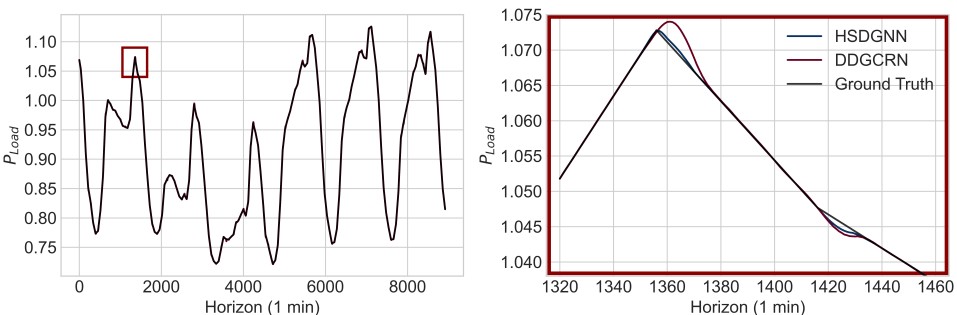

Figure 10: Prediction results given by HSDGNN and DDGCRN on the PSML dataset.

