# OpenReview forum: "Graph Neural Networks for Multivariate Time-Series Forecasting via Learning Hierarchical Spatiotemporal Dependencies"
_ICLR.cc/2024/Conference — Submitted to ICLR 2024_

### Official Review · Reviewer_MDiL · 2023-10-27

**Soundness:** 2 fair
**Presentation:** 2 fair
**Contribution:** 1 poor
**Rating:** 3
**Confidence:** 4

**Summary:**

The paper introduces HSDGNN, a neural network architecture for multivariate time series forecasting. HSDGNN propagates representations along both nodes and features using dynamic adjacency matrices, which can be considered analogous to spatial attention scores. The model uses static learnable embeddings for both time steps and nodes and other operators to condition the forecasts. Empirical results show performance competitive w.r.t. the state of the art.

**Strengths:**

* Decent empirical results.
* The ablation study is appreciated.
* Figures do a good job of clarifying aspects of the method.

**Weaknesses:**

The methodological novelty of the paper is quite limited and the selected datasets appear inappropriate.

* The main novelty of the paper consists of combining intra and inter-node graph convolutions on a dynamically learned adjacency matrix (with no constraint on the sparsity), which can be seen as a matrix of dynamic attention coefficients. However, several methods perform attention across different dimensions (e.g., [1]) and there's little novelty in the mechanism used to learn such matrices end-to-end.
* The architecture appears overly complicated compared to alternatives from the literature and such complexity is poorly justified (see minor comments).
* The considered datasets have at most 3 channels which does not justify modeling intra-channel dependencies with a graph. As shown by the ablation study in Tab. 2, removing the intra-dependency learning module results in a decrease in performance that is not statistically significant by considering the reported standard deviations.
* Empirical results on the existing traffic datasets are worse than those of much simpler baselines such as [2], which consists of a simple MLP with spatial and temporal embeddings.

Minor comments:

* Several claims are not supported by appropriate evidence. For example, in the introduction: "These methods can provide acceptable results depending on circumstances, but they may also introduce a higher degree of uncertainty, which can impact the reliability of predictions"; it is not clear what uncertainty the sentence is referring to and, at same time, how the introduced method should reduce uncertainty.
* "Even so, the temporal correlations regarding the changing spatial dependencies are not well-considered, which makes these methods ineffective in learning from dynamic graph topologies." There are a few papers that learn dynamic adjacency matrices using a mechanism similar to the one used here (e.g., [3]), but it is not clear how HSDGNN should be better than the current approaches.

Given the above, I cannot recommend acceptance.


[1] Ma et al. "CDSA: Cross-Dimensional Self-Attention for Multivariate, Geo-tagged Time Series Imputation" arxiv 2019\
[2] Shao et al. "Spatial-Temporal Identity: A Simple yet Effective Baseline for Multivariate Time Series Forecasting" CIKM 2022\
[3] Liu et al. "Multivariate Time-Series Forecasting with Temporal Polynomial Graph Neural Networks" NeurIPS 2022\

**Questions:**

Please clarify the novelty of the proposed method and justify this choice of benchmarks.

---

> ### Author Response · Authors · 2023-11-16
> **Responses to Reviewer MDiL**
>
> The authors would also like to thank you for your recognition of our work as well as the time and effort you have put into carefully reviewing our paper. In what follows, you can find point-by-point responses to your comments. Please feel free to contact us if there is any further problem.
>
> ***
> Q1. Novelty of the proposed method.
>
> The novelty of this paper mainly lies in that we proposed a framework for learning **dynamic** dependencies across different dimensions, i.e., temporal, spatial, and intra-node. Works such as [1] try to learn a fixed attention matrix from data, which is equivalent to executing the same transformation operation at each timestamp after the matrix is learned. However,  following the same transformation operation can not fully exploit the dependencies at different levels, as they can all vary with respect to time and location. On the other hand, works such as [3] and DDGCRN (SOTA) integrate the idea of learning dynamic dependencies, but they only consider dependency modeling along the temporal and spatial dimensions. We for the first time proposed a framework to model the dynamic hierarchical spatiotemporal dependencies in an effective and efficient manner.
>
> ***
> Q2. Complexity.
>
> In the proposed method, we introduce a novel framework that integrates several modules for multivariate time-series forecasting. Although the combination of these modules increases the complexity of the model, we prove in the experiments that such a combination can lead to better prediction performance. Moreover, we can see from Table 2 and Table 8 that the proposed method can achieve a higher prediction accuracy without compromising model scalability. The actual running time of our method is lower than that of DDGCRN. In summary, we can conclude that the proposed model can lead to better model performance without sacrificing time and space complexity.
>
> ***
> Q3. The considered datasets have at most 3 channels which does not justify modeling intra-channel dependencies with a graph. As shown by the ablation study in Tab. 2, removing the intra-dependency learning module results in a decrease in performance that is not statistically significant by considering the reported standard deviations.
>
> We adopt those widely used datasets to ensure fair comparisons with state-of-the-art works. These cases can be regarded as illustrations to directly prove the effectiveness of learning a hierarchical graph. Besides, the advantage of introducing the Intra-Dependency Learning Module can vary on different datasets. The absence of the IDLM can cause a performance decrease of up to 2.9% compared to the full model. Considering that the overall improvement of the proposed method over the state-of-the-art method DDGCRN is approximately 10-15%, the IDLM contributes to a relatively large portion of this improvement.
>
> ***
> Q4. Empirical results on the existing traffic datasets are worse than those of much simpler baselines such as [2], which consists of a simple MLP with spatial and temporal embeddings.
>
> MLP-based baselines such as STID [2] provide an efficient way for multivariate time-series modeling and prediction. However, the performance of such methods is inferior to the state-of-the-art GNN-based and Transformer-based methods, such as DDGCRN and PatchTST. In this regard, we further compare the proposed method with STID on the PEMSD4 and PEMSD8 datasets, which have also been used in the original paper. We train the STID model 10 times based on the code released by the author and report the average performance with standard deviations in Table r1. We can see from the results that HSDGNN still outperforms STID in most cases.
>
> Table r1. Comparisons of the performance of HSDGNN and STID on PEMSD4 and PEMSD8 datasets.
> ***
> |  METHOD  |  | D4 | | | D8 | |
> |  ----  | ----  | ----  | ----  | ----  | ----  | ----  |
> | HSDGNN | **18.254** | 30.443 | **12.065** | **13.718** | **23.405** | **9.077** |
> | | (0.064) | (0.132) | (0.177) | (0.154) | (0.137) | (0.136) |
> | STID | 18.752 | **30.235** | 13.085 | 14.458 | 23.465 | 9.598 |
> | | (0.027) | (0.041) | (0.185) | (0.028) | (0.056) | (0.078) |
>
> ***
> Q5. Several claims are not supported by appropriate evidence.
>
> We have made the following changes to our previous claims and provided references to support the claims.
>
> *Original 1*: These methods can provide acceptable results depending on circumstances, but they may also introduce a higher degree of uncertainty, which can impact the reliability of predictions. \
> *New 1*: However, the future values of the target attribute (e.g. traffic flow) not only depend on its historical records, but also on the influences of other related attributes. Neglecting such influences can significantly lower the prediction accuracy [Han 2021].
>
> References: [1] Han L, Du B, Sun L, et al. Dynamic and multi-faceted spatio-temporal deep learning for traffic speed forecasting[C]//Proceedings of the 27th ACM SIGKDD conference on knowledge discovery & data mining. 2021.

---

> > ### Comment · Reviewer_MDiL · 2023-11-17
> >
> > I want to thank the authors for the rebuttal and comments. My concerns about the novelty of the proposed architecture and the significance of the empirical results remain. I'll keep my score.

---

### Official Review · Reviewer_7UwB · 2023-10-31

**Soundness:** 3 good
**Presentation:** 3 good
**Contribution:** 2 fair
**Rating:** 5
**Confidence:** 4

**Summary:**

This paper primarily improves two main shortcomings of previous STGNN models: 1) the lack of consideration of dependencies among attributes of variables. 2) insufficient consideration of the temporal correlations in dynamic spatial dependencies.
To address these two shortcomings, this paper proposes a model called HSDGNN for multivariate time-series forecasting problems. HSDGNN captures the spatiotemporal dependencies among variables at two levels: attribute and variable. It also utilizes an additional temporal learning module to capture the temporal correlations among dynamic graph topologies.
Experiments demonstrate that the proposed model achieves performance improvement compared to the state-of-the-art STGNN models, without increasing the model size.

**Strengths:**

1.	Originality:
This paper explicitly establishes dependencies for each attribute of the variable, which is a departure from previous work that either combines multiple attributes together or does not consider the dependencies between attributes. This paper provides a valuable reference for future research in this area.
2.	Quality:
2.1	The paper's experiments are thorough and comprehensive. It demonstrates superior performance compared to the state-of-the-art STGNN models. Furthermore, the experiments confirm that while improving performance, the proposed model does not significantly increase training and testing time, as well as the model size.
2.2	The ablation experiments are well-executed, demonstrating the necessity of each module and providing reasonable explanations for the experiment results.
2.3	In the ablation experiments, HSDGNN_w/o_MF outperformed DDGCRN, showcasing the superiority of the model framework. Furthermore, incorporating the remaining attributes resulted in even higher performance, underscoring the importance of establishing attribute dependencies.
2.4	The model exhibits remarkable stability, as its performance does not significantly fluctuate with parameter variations.
3.	Clarity:
Overall, the paper is clearly written and well organized. Figure 2 clearly illustrates the overall workflow and the functions of each module in the model.
4.	Significance:
In my view, the major contribution of this paper is the modeling of attribute dependencies among variables, which introduces a fresh perspective and a good direction for future research. Future work can conduct more in-depth research on how to better explore the dependencies between attributes.

**Weaknesses:**

1.	In this paper, only the main attribute (traffic flow) is predicted, and the other attributes served as auxiliary attributes for the prediction of the main attribute. If only the main attribute is predicted, the other attributes can be considered as providing additional useful information for the prediction of the main attribute. However, this does not necessarily reflect the presence of inherent dependencies between attributes, as each attribute can be treated as the main attribute.
2.	In Figure 2, the Intra-dependency Tensor (I) should be a 3x3 grid instead of a 4x4 grid.
3.	In Appendix A.2, HSDGNN is incorrectly written as MSDGNN.

**Questions:**

1.	For weakness1. Although the ablation experiments suggest that considering only the performance of the main attribute can surpass that of DDGCRN, this only demonstrates the superiority of the model architecture. Therefore, I hope the authors can provide the prediction results and ablation experiments for the other attributes (occupancy, speed).
2.	In long-term prediction, is the horizon of input also 12? If the horizon of input is 12, why are the values of PEMSD8 in Figure7 and Figure6 slightly different under the first 12 timestamps? If the horizon of input is not 12, can author indicate horizon of input of HSDGNN and DDGCRN for long-term prediction performance in the paper?
3.	I am not sure whether the horizon of input of HSDGNN and DDGCRN is the same in long-term prediction. If not, can the author modify the experiment to compare the performance of HSDGNN and DDGCRN under the same horizon of input?

---

> ### Author Response · Authors · 2023-11-16
> **Responses to Reviewer 7UwB**
>
> The authors would also like to thank you for your recognition of our work as well as the time and effort you have put into carefully reviewing our paper.  In what follows, you can find point-by-point responses to your comments. Please feel free to contact us if there is any further problem.
> ***
> Q1. In this paper, only the main attribute (traffic flow) is predicted, and the other attributes serve as auxiliary attributes for the prediction of the main attribute. If only the main attribute is predicted, the other attributes can be considered as providing additional useful information for the prediction of the main attribute. However, this does not necessarily reflect the presence of inherent dependencies between attributes, as each attribute can be treated as the main attribute. Although the ablation experiments suggest that considering only the performance of the main attribute can surpass that of DDGCRN, this only demonstrates the superiority of the model architecture. Therefore, I hope the authors can provide the prediction results and ablation experiments for the other attributes (occupancy, speed).
>
> Thanks for the suggestion. Following this idea, we modify the output layers of both DDGCRN and the proposed HSDGNN to enable them to generate predictions for all the attributes at the same time. We re-train the new models on the PEMSD8 dataset, and calculate the three evaluation metrics (MAE/RMSE/MAPE) for each attribute separately. In this way, we can quantify the influence of intra-dependency modeling when predicting all attributes. The results can be seen in Table 1, where we also include the ablation study results.
>
> **Table 1. Prediction results on traffic flow (S), occupancy (O) and speed (S)**
> ***
> | METHOD  |  | F |  |  | O |  |  | S |  |
> | ---- | ---- | ---- | ---- | ---- | ---- | ---- | ---- | ---- | ---- |
> | METHOD | MAE | RMSE | MAPE | MAE | RMSE | MAPE | MAE | RMSE | MAPE |
> | DDGCRN | 14.746	| 24.317 | 9.773 | 0.012 | 0.027 | 19.951 |1.910 | 4.788 | 4.652 |
> | HSDGNN | **13.625** | **23.310** | **8.911** | **0.008** | **0.021** | **14.655** | **1.332** | **3.254** | **2.946** |
> | HSDGNN_w/o_IDLM | 13.824 | 24.253	| 9.071 | 0.010 | 0.022 | 16.268 | 1.370 | 3.322 | 3.054 |
> | HSDGNN_w/o_GRU1 | 13.670 | 23.530 | 9.036 | 0.010 | 0.022 | 17.026 | 1.367 | 3.342 | 3.056 |
> | HSDGNN_w/o_GRU2 | 14.417 | 23.720 | 9.654 | 0.009 | 0.023 | 17.848 | 1.367 | 3.286 | 2.981 |
> | HSDGNN_w/o\_DG | 14.807 | 23.903 | 9.829 | 0.013 | 0.027 | 18.842 | 1.366 | 4.370 | 3.074 |
>
> We can draw several conclusions from the results: 1) Compared to DDGCRN, the proposed method can improve the prediction accuracy of all attributes; 2) The absence of any component will decrease model performance, either regarding the main attribute or the auxiliary attributes; 3) The results also prove the necessity of modeling the inherent dependencies between attributes.
>
> ***
> Q2. In long-term prediction, is the horizon of input also 12? If the horizon of input is 12, why are the values of PEMSD8 in Figure7 and Figure6 slightly different under the first 12 timestamps? If the horizon of input is not 12, can author indicate horizon of input of HSDGNN and DDGCRN for long-term prediction performance in the paper?
>
> The horizon of the input is also 12 for long-term prediction. The results are slightly different because we have to modify the size of the output layer of both DDGCRN and HSDGNN to generate 72 steps of predictions. In other words, we re-train two new models for this task, so the results can be slightly different.
>
> ***
> Q3. I am not sure whether the horizon of input of DDGCRN and HSDGNN is the same in long-term prediction. If not, can the author modify the experiment to compare the performance of HSDGNN and DDGCRN under the same horizon of input?
>
> We adopt the same input length of 12 for DDGCRN and HSDGNN when performing long-term prediction.

---

> > ### Comment · Reviewer_7UwB · 2023-11-22
> > **Acknowledgement of authors' response**
> >
> > Thank the authors for the response. After carefully reading all other reviewers' comments and rebuttals, I would like to keep my original score.

---

### Official Review · Reviewer_MTEy · 2023-11-01

**Soundness:** 3 good
**Presentation:** 3 good
**Contribution:** 2 fair
**Rating:** 3
**Confidence:** 4

**Summary:**

This paper introduces a hierarchical spatial-temporal dependency graph neural networks (HSDGNN) for multi-variate time series forecasting. The proposed HSDGNN has two levels. At the first level, for each multi-variate time series collected at a sensory station, HSDGNN learns the dependencies (graph) of different variates, and then use a GRU to model temporal dynamics of each station. At the second level, HSDGNN learns a dynamic spatial-temporal graph among different sensory stations and then use another GRU to model the temporal dynamics of the entire graph. The model is evaluated on several benchmark datasets and the proposed method could outperform baselines.

**Strengths:**

1. The proposed model is technically sound. The graph learning, spatial modeling and temporal modeling are based on prior works, and thus the proposed HSDGNN should have a good performance.
2. The writing of the paper is clear and easy to follow.
3. The code is provided.

**Weaknesses:**

1. The contribution of this paper is somewhat incremental. The proposed method is kind of simple combination of existing methods [1][2][3].
2. According to the results of ablation study (Table 3), it seems that IDLM (the intra graph) is less effective. The performance of the full model HSDGNN and the ablated version HSDGNN_w/o_IDLM does not have a significant difference.
3. How will your model perform if you simply replace IDML with some other simple static graphs (e.g., the graph constructed by Pearson Correlation or other prior knowledge)?

[1] Han, Liangzhe, et al. "Dynamic and multi-faceted spatio-temporal deep learning for traffic speed forecasting." Proceedings of the 27th ACM SIGKDD conference on knowledge discovery & data mining. 2021.

[2] Yaguang Li, Rose Yu, Cyrus Shahabi, and Yan Liu. Diffusion convolutional recurrent neural network: Data-driven traffic forecasting. In International Conference on Learning Representations (ICLR ’18), 2018.

[3] Lei Bai, Lina Yao, Can Li, Xianzhi Wang, and Can Wang. Adaptive graph convolutional recurrent network for traffic forecasting. Advances in neural information processing systems, 2020.

**Questions:**

Please refer to weaknesses.

---

> ### Author Response · Authors · 2023-11-17
> **Responses to Reviewer MTEy**
>
> The authors would also like to thank you for your recognition of our work as well as the time and effort you have put into carefully reviewing our paper.  In what follows, you can find point-by-point responses to your comments. Please feel free to contact us if there is any further problem.
>
> ***
> Q1.The contribution of this paper is somewhat incremental. The proposed method is kind of simple combination of existing methods.
>
> The proposed method benefits from these works, however, it is not a simple stack of them. We adopt a similar backbone as that in [1], but on top of that, we introduce the modeling of dependency and dynamics at different levels in a unified framework, especially the modeling of intra-dependences among attributes. As we have demonstrated through the experiment, these designs can improve prediction accuracy by a great margin. Besides, compared to the work [2] which also considers the effects of additional attributes, our method can maintain the model scalability given a consistent model size regarding the number of variables.
>
> References
> [1] Weng W, Fan J, Wu H, et al. A Decomposition Dynamic graph convolutional recurrent network for traffic forecasting[J]. Pattern Recognition, 2023, 142: 109670.
> [2] Han L, Du B, Sun L, et al. Dynamic and multi-faceted spatio-temporal deep learning for traffic speed forecasting[C]//Proceedings of the 27th ACM SIGKDD conference on knowledge discovery & data mining. 2021: 547-555.
>
> ***
> Q2. According to the results of ablation study (Table 3), it seems that IDLM (the intra graph) is less effective. The performance of the full model HSDGNN and the ablated version HSDGNN_w/o_IDLM does not have a significant difference.
>
> The advantage of introducing the Intra-Dependency Learning Module can vary on different datasets. The absence of the IDLM can cause a performance decrease of up to 2.9% compared to the full model. Considering that the overall improvement of the proposed method over the state-of-the-art method DDGCRN is approximately 10-15%, the IDLM contributes to a relatively large portion of this improvement.
>
> ***
> Q3. How will your model perform if you simply replace IDML with some other simple static graphs (e.g., the graph constructed by Pearson Correlation or other prior knowledge)?
>
> The IDLM was designed to capture time-varying dependencies among different attributes of a variable. To further demonstrate the necessity of IDLM, we replace it with a static graph which is learned from data for comparison. Specifically, we initialize the intra-dependency values among attributes by a 3*3 random matrix, which is also trained in an end-to-end manner. After training, we can obtain the static graph that reflects the fixed correlations among attributes. We train the new model (HSDGNN_w_s) on each dataset 10 times and report the average results and deviations in Table r1.
>
> Table r1. Comparisons between HSDGNN and HSDGNN_w_s on all datasets
> ***
> | METHOD | | D4 | | | D5 | | | D8 | | | D11 | | | PS | |
> |----|----|----|----|----|----|----|----|----|----|----|----|----|----|----|----|
> | HSDGNN | 18.254 | 30.443 | 12.065 | 10.414 | 16.526 | 15.387 | 13.718 | 23.405 | 9.077 | 13.586 | 22.627 | 13.706 | 0.245 | 0.649 | 25.632|
> | | (0.064) | (0.132) | (0.177) | (0.115) | (0.313) | (0.286) | (0.154) | (0.137) | (0.136) | (0.025) | (0.058) | (0.086) | (0.025) | (0.067) | (3.740) |
> | HSDGNN_w_s  | 18.331| 30.942 | 12.109 | 10.470 | 16.659 |15.644 | 13.827 | 23.513 | 9.108 | 13.622 | 22.729 | 13.875 | 0.313 | 0.695 | 32.786 |
> | | (0.048) | (0.282) | (0.052) | (0.059) | (0.120) | (0.163) | (0.075) | (0.082) | (0.048) | (0.025) | (0.051) | (0.096) | (0.020) | (0.086) | (2.155) |
>
> We can see from the results that HSDGNN_w_s performs worse than the original HSDGNN method in all cases. It proves that the intra-dependencies among attributes also vary with time, and our model can benefit from modeling such dynamic dependencies using IDLM.

---

> > ### Comment · Reviewer_MTEy · 2023-11-19
> >
> > Thank you for your responses. I've read the responses and I'd like to keep my score.

---

### Official Review · Reviewer_kfob · 2023-11-01

**Soundness:** 2 fair
**Presentation:** 1 poor
**Contribution:** 1 poor
**Rating:** 3
**Confidence:** 4

**Summary:**

The paper introduces a GNN-based architecture for multivariate time series forecasting where the relational information among the time series - the graph used by the GNN - is not given, but learned from data. In particular, the proposed method is designed for learning a dynamic graph which is structured into two levels to overcome limitations of previous work.

**Strengths:**

The ideas of constructing a hierarchical and dynamic graph are interesting and worth exploring. The empirical analysis reports improved performance with respect to the considered baselines.

**Weaknesses:**

My major concerns are the following:
- Several claims appear unsubstantiated, especially in the fourth paragraph of the introduction. References and/or pieces of evidence to support such claims should be provided.
- The technical contribution is limited. The proposed method appears as a minor modification of existing works (e.g., Bai et al., 2020, Han et al., 2021, Weng et al., 2023).

**Questions:**

I am not fully convinced by the experimental setting. What is the relevance of learning a hierarchical graph in the considered setting where each location (variable in the paper terminology) has only three components? Fig 9a shows that the graph of attributes is a fully connected one, with all edge weights ~1. Could you clarify what is used to replace the IDLM in `HSDGNN_w/o_IDLM` and the dynamic graph in `HSDGNN_w/o_DG`?

Confusing notation
- Sets and tensors appear to be used interchangeably, the tensor dimensionality is rarely provided.
- Sec. 3.1 defines $\mathcal G_s$ as both a set and a graph.
- Eq. 6: $\mathbf W_{G_1}$ appear undefined,
- Eq. 6: notation $\mid \mathbf W_{G_1}$ is undefined
- Before Eq. 7: notation $\boldsymbol T_e(\mathbf X)$ is undefined

---

> ### Author Response · Authors · 2023-11-16
> **Responses to Reviewer kfob**
>
> The authors would also like to thank you for your recognition of our work as well as the time and effort you have put into carefully reviewing our paper.  In what follows, you can find point-by-point responses to your comments. Please feel free to contact us if there is any further problem.
>
> ***
> Q1. Several claims appear unsubstantiated, especially in the fourth paragraph of the introduction. References and/or pieces of evidence to support such claims should be provided.
>
> We have made the following changes to our previous claims and provided references to support the claims.
>
> Original 1: These methods can provide acceptable results depending on circumstances, but they may also introduce a higher degree of uncertainty, which can impact the reliability of predictions.
> New 1: However, the future values of the target attribute (e.g. traffic flow) not only depend on its historical records, but also on the influences of other related attributes. Neglecting such influences can significantly lower the prediction accuracy [Han 2021].
>
> Original 2: Even so, the temporal correlations regarding the changing spatial dependencies are not well-considered, which makes these methods ineffective in learning from dynamic graph topologies.
> New 2: Although these methods construct a dynamic graph at each timestamp, the temporal correlations among these dynamic graphs are not well-considered, which makes these methods ineffective in learning from dynamic graph topologies.
>
> References:
> [1] Han L, Du B, Sun L, et al. Dynamic and multi-faceted spatio-temporal deep learning for traffic speed forecasting[C]//Proceedings of the 27th ACM SIGKDD conference on knowledge discovery & data mining. 2021: 547-555
>
> ***
> Q2. The technical contribution is limited. The proposed method appears as a minor modification of existing works (e.g., Bai et al., 2020, Han et al., 2021, Weng et al., 2023).
>
> The proposed method benefits from these works, however, it is not a simple stack of them. We adopt a similar backbone as that in [Weng, 2023], but on top of that, we introduce the modeling of dependency and dynamics at different levels in a unified framework, especially the modeling of intra-dependences among attributes. As we have demonstrated through the experiment, these designs can improve prediction accuracy by a great margin. Besides, compared to the work [Han 2021] which also considers the effects of additional attributes, our method can maintain the model scalability given a consistent model size regarding the number of variables.
>
> ***
> Q3. I am not fully convinced by the experimental setting. What is the relevance of learning a hierarchical graph in the considered setting where each location (variable in the paper terminology) has only three components? Fig 9a shows that the graph of attributes is a fully connected one, with all edge weights ~1.
>
> It depends on the datasets. We adopt those widely used datasets to ensure fair comparisons with state-of-the-art works. These cases can be regarded as illustrations to directly prove the effectiveness of learning a hierarchical graph. Besides, we did not normalize the attention matrix such that the values can range from 0 to approximately 3.8 after the training of the model (considering the PEMSD8 dataset). In the case shown in Fig 9, the edge weights corresponding to that specific period are within the range of 0.8-1.3.
>
> ***
> Q4. Could you clarify what is used to replace the IDLM in HSDGNN_w/o_IDLM and the dynamic graph in HSDGNN_w/o_DG?
>
> For HSDGNN_w/o_IDLM, we exclude the intra-dependency learning process in the original design by using the raw input X as the input for the Temporal-dependency Learning module, i.e., X -> Attribute Fusion F, as shown in Figure 2.
>
> For HSDGNN_w/o_DG, we adopt a consistent graph to model the spatial dependency among different variables. To achieve this, we multiply the Static Node Embedding (M)  by its transpose to obtain an attention matrix, which is then broadcast to all timestamps. In other words, we do not consider the dynamic dependencies among multiple variables in this case.
>
> ***
> Q5. Confusing notation
>
> We have made some modifications in the manuscript to clarify these confusing terms.

---

> > ### Comment · Reviewer_kfob · 2023-11-17
> > **Thanks**
> >
> > I thank the authors for the clarifications. I have read the rebuttal and the comments made by the other reviewers. I would like to keep my original score.

---

### Meta-Review · Area_Chair_15Sg · 2023-12-08

**Metareview:**

The paper addresses interesting spatiotemporal problems with GNNs. However, none of the reviewers is really in favour of accepting the paper. They all highlight problems with novelty and significance in their reviews and consider the proposed approach straight forward given related work.

**Justification For Why Not Higher Score:**

The technical contribution appears straight forward and a combination of existing methods. There are many minor issues raised in the reviews as well.

**Justification For Why Not Lower Score:**

N/A

---

### Decision · Program_Chairs · 2024-01-16

Reject